# Research on a Precision Calibration Model of a Flexible Strain Sensor Based on a Variable Section Cantilever Beam

**DOI:** 10.3390/s23104778

**Published:** 2023-05-16

**Authors:** Qi Wang, Jianjun Cui, Yanhong Tang, Liang Pang, Kai Chen, Baowu Zhang

**Affiliations:** 1School of Information Science and Engineering, Zhejiang Sci-Tech University, Hangzhou 310018, China; 2National Institute of Metrology, Beijing 100029, China; 3Metrology and Testing Institute of Tibet Autonomous Region, Lhasa 850000, China; 4College of Metrology and Measurement Engineering, China Jiliang University, Hangzhou 310018, China

**Keywords:** flexible sensor, variable section cantilever beam, large deflection, strain, calibration model

## Abstract

The flexible strain sensor’s measuring range is usually over 5000 με, while the conventional variable section cantilever calibration model has a measuring range within 1000 με. In order to satisfy the calibration requirements of flexible strain sensors, a new measurement model was proposed to solve the inaccurate calculation problem of the theoretical strain value when the linear model of a variable section cantilever beam was applied to a large range. The nonlinear relationship between deflection and strain was established. The finite element analysis of a variable section cantilever beam with ANSYS shows that the linear model’s relative deviation is as high as 6% at 5000 με, while the relative deviation of the nonlinear model is only 0.2%. The relative expansion uncertainty of the flexible resistance strain sensor is 0.365% (k = 2). Simulation and experimental results show that this method solves the imprecision of the theoretical model effectively and realizes the accurate calibration of a large range of strain sensors. The research results enrich the measurement models and calibration models for flexible strain sensors and contribute to the development of strain metering.

## 1. Introduction

Flexible strain sensors [1,2] are made of flexible materials based on the principle of the resistance strain effect or other strain sensing effects. With the advantages of strong flexibility and good tensile performance [3], it enables the sensing and measurement of large strain ranges and can be used in many emerging technology fields such as robot tactile sensing [4], human motion detection equipment [5,6], telemedicine [7,8,9], and so on.

At present, the accuracy of flexible strain sensor’s strain sensing needs to be improved, especially in the accurate measurement of strain characteristics, including calibration models, measurement methods, and accuracy evaluation. The main reason is that the deformation range of the flexible strain sensor (5000 με) [10] exceeds the applicable range of the conventional strain calibration method (1000 με) [11], and the flexible strain sensor can sense the strain in at least two dimensions [12], while the conventional strain sensor is usually linear strain sensing in one dimension. These characteristics put forward new requirements for the strain sensing characteristics calibration. However, the conventional strain calibration method [13,14,15,16] has important reference values for the development of flexible strain sensor calibration technology. For example, the variable section cantilever beam can produce strain when it is deformed due to the force, which can be used for the calibration of conventional strain sensors [17]. Because it can produce uniform surface strain, it is possible to calibrate the sensing characteristics of the flexible strain sensor by using a variable section cantilever beam.

When the variable section cantilever beam was initially used for strain calibration, the loading method [18] was adopted to determine the surface strain value of the beam by the density of the load. However, the elastic modulus of the model parameter cannot be ascertained, which leads to the strain not being traced. Hou et al. [19] proposed to calculate the surface strain of the beam by measuring the deflection, thus transforming the problem of strain measurement into the problem of deflection measurement, and the deflection can be traced to the source through displacement, thus solving the traceability problem of strain measurement.

However, the econometric model makes the approximate assumption of small deflection deformation when solving the differential equation of deflection, thus introducing the model error. Zhao et al. [20] analyzed the large deflection angle of a variable section cantilever beam and obtained the exact solution by transforming the differential equation of deflection into a transcendental equation. They concluded that the relative error between the approximate solution without considering the deflection angle and the exact solution would increase with the deflection growth. Polilov et al. [21] simplified the cantilever beam as a sheet spring with accumulated elastic energy, analyzed the stiffness, section size ratio, shear force, and other aspects, and determined the parameter range that could ignore the influence of the deflection angle.

It can be seen that when the bending degree of a variable section cantilever beam is large, the conventional deflection measurement model is not accurate enough and is difficult to apply to the calibration of a flexible strain sensor.

Yan et al. [22] adopted analytical methods such as finite element analysis and measurement experiments to solve the problems existing in the measurement model. By modifying the conventional deflection method measurement model, they obtained the engineering measurement application model. However, this model has different correction functions for different sizes of variable section cantilevers, which makes it difficult to apply, and the parameter values are not universal.

Therefore, in order to meet the need for a flexible strain sensor’s quantitative traceability, this paper deduces the model, which can be analyzed precisely, to solve the problem of inaccurate strain calculation of a variable section cantilever beam with large deflection deformation, and then verifies the science and rationality of the measurement model through a finite element simulation digital experiment. Compared with the analysis results of the conventional deflection method, the strain calculation result of this model is more accurate. The calibration experiments of the resistance strain sensor packaged with a flexible substrate are carried out, and the sensing characteristic parameters of the flexible strain sensor are obtained.

## 2. Measurement Model

### 2.1. The Relationship between Deflection and Strain of a Variable Section Cantilever Beam

The variable section cantilever beam is an equal-thickness triangular cantilever beam with a fixed bottom end and a free tip end. When the load is applied at the free end, the same strain values are on the surface of the variable section cantilever beam. Its model is shown in Figure 1.

The length of the variable section cantilever beam is denoted as *L*, the thickness is denoted as *h*, the width of the fixed end is denoted as *b*, and *x* is set as the distance from one point on the beam to the fixed end. According to the principle of material mechanics [23], the variable section cantilever beam is bent by applying load *F* at the loading point of the free end. Due to the small amount of deformation caused by shear force, it is able to be ignored. Moreover, assuming that the bending of the variable section cantilever beam is an arc section, the differential equation of the deflection line is as follows [24]:(1)$$\frac{\frac{d^{2}y(x)}{dx^{2}}}{{\left(1+{\left(\frac{dy(x)}{dx}\right)}^{2}\right)}^{\frac{3}{2}}}=\frac{12FL}{Ebh^{3}},$$
where *y*(*x*) is the deflection value of the variable section cantilever beam at point *x*, and *E* is the elastic modulus of the variable section cantilever beam.
(2)$$f(F)=\frac{12FL}{Ebh^{3}},$$

The relation between the surface strain value and the load value of the variable section cantilever beam is the following [25]:(3)$$\varepsilon =\frac{6FL}{Ebh^{2}},$$

Combined with Equations (1) and (3), the direct relationship between deflection and strain can be established:(4)$$\varepsilon =\frac{\frac{d^{2}y(x)}{dx^{2}}}{{\left(1+{\left(\frac{dy(x)}{dx}\right)}^{2}\right)}^{\frac{3}{2}}}\frac{h}{2},$$

The fixed end of the variable section cantilever beam has no deformation, and its deflection and deflection value are 0, so the boundary condition is as follows:(5)$$\frac{dy(x)}{dx}{|}_{x=0}=0,$$
(6)y(0)=0,

By solving the differential equation, the expression of the variable section cantilever beam’s deflection line and the surface strain of the variable section cantilever beam can be obtained.

### 2.2. Conventional Small Deflection Strain Analysis Model

When the bending degree of the variable section cantilever beam is small, the deflection angle of the beam is approximately 0, so the denominator of the left end of Equation (1) is approximately 1.
(7)$$\frac{d^{2}y(x)}{dx^{2}}=f(F),$$

Then *x* is continuously integrated twice at both ends, and the expression *y*(*x*) of the deflection line of the variable section cantilever beam is obtained by substituting the boundary condition Equations (5) and (6).
(8)$$y(x)=\frac{f(F)x^{2}}{2},$$

Combined with Equations (2), (3) and (8), the relation between surface strain and deflection of the variable section cantilever beam is the following:(9)$$\varepsilon^{\prime }=\frac{hy(x)}{x^{2}},$$

This model (hereinafter referred to as the linear model) simplifies the solving process of the deflection’s differential equation by ignoring the deflection angle, but introduces the theoretical error, which affects the strain calculation results, and cannot be applied to calculate strain value under the condition of large deflection. Therefore, it cannot meet the large deflection measurement requirements of a flexible strain sensor.

### 2.3. Flexible Large Deflection Strain Analysis Model

The measuring range of the flexible strain sensor is much larger than that of the conventional strain sensor. When the full range calibration is carried out by the variable section cantilever beam, the bending degree of the beam is larger, and the deflection angle parameter cannot be ignored. In order to improve the precision of theoretical calculations and make the calculation result of theoretical strain more accurate, it is necessary to obtain the exact analytical formula of deflection. The specific process is as follows:(10)$$\tan u=\frac{dy(x)}{dx},$$

(1), (2) and (10) are combined to obtain the following:(11)$$\cos u\frac{du}{dx}=f(F),$$

Equation (11) is integrated with respect to *x* at both ends, and the boundary conditions (5) are substituted:(12)sinu=f(F)x,

Combining Equations (12) and (10), which can be solved as follows:(13)$$\frac{dy(x)}{dx}=\frac{f(F)x}{\sqrt{1-f{\left(F\right)}^{2}x^{2}}},$$

By integrating *x* at both ends of Equation (13) and substituting boundary conditions (6), the flexural equation of the variable section cantilever beam can be written as follows:(14)$$y(x)=\frac{1-\sqrt{1-f{\left(F\right)}^{2}x^{2}}}{f(F)},$$

By combining Equations (1), (2), (4) and (14), the relation between surface strain and deflection of the variable section cantilever beam can be written as follows:(15)$$\varepsilon =\frac{hy(x)}{x^{2}+y{\left(x\right)}^{2}},$$

The relationship between strain and deflection in the model is nonlinear (hereinafter referred to as the nonlinear model). With Equation (15) as the reference value, the relative deviation introduced by Equation (9) due to ignoring the deviation angle is as follows:(16)$$e=\frac{y^{2}(x)}{x^{2}},$$

The relative deviation is related to the position of the deflection measuring point and the deflection measuring value. Under the condition that the position of the measuring point remains unchanged, the deflection value of the cantilever beam increases with the increase of the load, and the relative deviation value of the calculated results of Equation (9) increases.

According to Formula (15), the surface strain value of the variable section cantilever beam can be calculated by measuring the location *x*, the deflection value of the section where the point *x* is located, and the thickness h of the variable section cantilever beam. All measured values can be traced to the displacement. The deflection measuring point of the nonlinear measurement model is located at the same location as the strain measuring point, which can realize the accurate calculation of strain in large deflection deformation and meet the need for calibration and traceability of flexible strain sensors.

## 3. Simulated Analysis

Finite element analysis divides geometric structures into elements and obtains the state solution of the structure by the approximate method. The simulation experiment of a variable section cantilever beam was carried out by ANSYS2022 R2 software to analyze the variation of deflection and surface strain distribution of a variable section cantilever beam, and the strain calculation results of the two models were compared and analyzed.

### 3.1. Characteristic Parameters of a Variable Section Cantilever Beam

SolidWorks was used to construct a three-dimensional model of a variable section cantilever beam. In order to facilitate the application of loads, the free end was widened and designed as a rectangular section. Since the surface strain of the beam is only related to the torque and modulus of the bending section, the design would not affect the surface strain value theoretically. The reference values of spring steel are used for the elastic modulus and density of the variable section cantilever beam model. Since Formula (15) does not include the two parameters, the setting of its value will not affect the verification of the nonlinear relationship between deflection and strain in the model.

The material of the variable section cantilever beam is 65 Mn spring steel. The loading point is the intersection of the two waist extension lines, 360 mm from the fixed end of the variable section cantilever beam. The structural model is shown in Figure 2. The dimension parameters and material characteristics are shown in Table 1.

According to Formula (3), the specific strain value is generated on the surface of the cantilever beam by changing the load value at the loading point after loading 100 N, 300 N, and 500 N successively, comparing and analyzing the calculation results of the two models with the simulation data as a reference.

### 3.2. Deflection Simulation Analysis

Firstly, the designed variable section cantilever beam model was meshed, then the geometric structure path was set along the axis, and the measurement points were set at equal intervals. Finally, a 100 N load was applied at the loading point, and the simulation values at the measurement points were recorded.

Figure 3 shows the overall deflection distribution of the variable section cantilever beam, and Figure 4 shows the simulation deflection curve and the theoretical deflection curve calculated by Equation (14). Due to the widening design of the free end, the modulus of the flexural section increases. As a result, the deflection line of the cantilever beam in the area 285–360 mm away from the fixed end does not meet the circular arc hypothesis, and the difference between the simulated deflection and the theoretical deflection calculated by Equation (14) exceeds 0.1 mm. Therefore, when measuring deflection, the measuring point is selected in the variable section area.

### 3.3. Strain Simulation Analysis

When a 100 N load is applied, the strain distribution on the surface of the variable section cantilever beam is shown in Figure 5. In the variable section area, the strain distribution on the beam’s surface is also uneven due to the influence of the fixed end support and the sudden change of beam width at 285 mm. At the fixed end, at the junction of the variable section and the widened section, the strain value changes greatly, so it is not suitable to measure the strain value.

The simulation deflection value was substituted into Equations (9) and (15), and the theoretical strain value was calculated. The calculated result’s relative deviation between the two theoretical models was analyzed with the simulation strain value as the reference. The simulation strain and theoretical strain calculation results are shown in Figure 6. In the variable section area, there is a deviation between the calculation results of the linear model and the simulation results. In the range of 120–240 mm from the fixed end, the simulation analysis results are consistent with the theoretical analysis data of Equation (15), which is the effective measurement area of the variable section cantilever beam.

The relative deviation between theoretical calculation results and simulation strain results is shown in Figure 7. Under 100 N load, the relative deviation of linear model results within the effective measurement area is less than 0.25%. With the increase in distance between the measuring point and the fixed end, the relative deviation of the calculated results also gradually increases. While the relative deviation of the calculated results of the nonlinear model strain is less than 0.02%.

Applying the 300 N load at the loading point, the simulation strain and theoretical strain calculation results are shown in Figure 8. The variation trend of the three groups of data is the same as that of the 100 N load simulation.

The relative deviation between the theoretical calculation results and the simulation strain results is shown in Figure 9. With the increase in load, the bending degree of the variable section cantilever beam increases, and the relative deviation of the linear model within the measurement area reaches 2%, which is no longer suitable for the calibration work of the flexible strain sensor.

Applying the 500 N load at the loading point, the simulation strain and theoretical strain calculation results are shown in Figure 10. The strain distribution on the surface of the beam changes obviously, and there are great differences in the strain values at different measuring points. The difference between the strain values calculated by the linear model and the simulated strain values increases further.

According to the analysis results in Figure 11, the relative deviation of the linear model reaches 6%, and the growth rate of this value increases with the increase of the deflection, which accords with the growth trend of Equation (16). However, in the range of 120–220 mm from the fixed end, the relative deviation of the proposed method is still within 0.2%. As the effective measurement area is reduced in the analysis of the 500 N simulation data, the part close to the fixed end in the effective measurement area can be pasted when the flexible strain sensor is calibrated and measured in a large range.

The error of the linear model is introduced when ignoring the deflection angle results in a large deflection theoretical value. According to the three sets of simulation data results, the error will increase with the increase of the distance from the measuring point to the fixed end at the same load and increase with the increase of the deflection at the same measuring point. The nonlinear model is proposed to obtain the exact solution of the flexural equation, and the simulation results agree with the theoretical values.

The deflection value and strain value are measured at a distance of 130 mm from the fixed end. The simulation data and theoretical strain calculation results under different loads are shown in Table 2, and the comparison results between simulation data and theoretical data are shown in Figure 12. With the increase in the deflection value, the difference between the calculated strain value of the linear model and the simulated strain value increases gradually. The calculated strain value of the nonlinear model is closer to the simulated strain value, and the maximum difference is 4.2 με.

## 4. Measurement Experiment

### 4.1. Equipment Design

The strain calibration device, as shown in Figure 13, mainly includes the main control computer, which is used for instrument control and data analysis; a high-precision deflection meter, with a measuring range of 12 mm and measuring accuracy of 2 μm, used to measure the change of deflection; a flexible resistance strain sensor for measuring the surface strain change of the beam; a seven-digit semi-digital multimeter for resistance measurement of the flexible resistance strain sensor; an M1-level weight for applying load; a stainless steel bracket used for fixing a variable section cantilever beam; and a deflection meter probe.

### 4.2. Experimental Design

#### 4.2.1. Experimental Scheme

The flexible resistive strain transducer was attached to the cantilever beam with its central axis coinciding with the central axis of the variable section beam and centered 138.30 mm from the fixed end. Fix the deflection meter head so that the measuring contact is perpendicular to the surface of the variable section cantilever beam, measure the distance from the contact position to the fixed end, and record. The distance between the position of the contact and the fixed end is measured and recorded. The experimental environment temperature is 20 °C, and the temperature change does not exceed 0.2 °C/h. Before the system is used, it is placed in the experimental environment for half an hour to balance the temperature.

At the beginning of the experiment, the variable section cantilever beam was preloaded, and the load was greater than the maximum load value of the experiment. Then, after withdrawing the load and waiting for the system to stabilize, the initial resistance value of the flexible resistance strain sensor was recorded, and the initial value indicated by the deflection meter was set as the reference value. At the free end of the variable section cantilever beam, the weight is suspended successively to impose loads. When the system is stabilized, the deflection value of the variable section cantilever beam and the resistance value of the flexible resistance strain sensor are recorded, respectively. The above procedure was repeated three times to complete the measurement of the experimental data.

#### 4.2.2. Experimental Measurement Data

When no weight is loaded, the initial resistance value of the flexible resistance strain sensor is 120.7222 Ω. The deflection measuring point of the variable section cantilever beam is 138.30 mm away from the fixed end. The variation of the deflection value measured by the deflection meter is calculated by Equation (15) to obtain the variation of the surface strain value of the variable section cantilever beam. The experimental data of three measurements are shown in Table 3, Table 4 and Table 5, and the linear fitting of the data is shown in Figure 14, Figure 15 and Figure 16. Data fitting correlation R^2^ ≥ 99.999%.

It can be seen that the theoretical strain calculation results of the linear model are larger than those of the nonlinear model from the three bidirectional measurements, and the difference between the two increases with the increase of the deflection. When the deflection of the measured point is 2.0229 mm, the theoretical strain error calculated by Equation (9) is 0.11 με, and the relative error is 0.0214%. In addition, it is necessary to pay attention to the anelasticity [26] of the variable section cantilever beam. After unloading, enough time should be left to wait for its deformation recovery. Otherwise, the measured values of deflection and strain will be larger under the same load, as shown in Table 3. However, it does not affect the experimental results because the bending deformation of the beam changes synchronously with the surface strain.

Generally, the deformation of the flexible sensor can reach more than 3000 με, and the deflection of the measuring point can reach 11.56 mm. According to Formula (9), the relative deviation has reached 0.7%. The relation between relative error and deflection introduced by Equation (9) is shown in Figure 17. With the increase in the deflection value, the relative error presents a trend of accelerating growth.

The calculation formula for the sensitivity coefficient of the flexible resistance strain sensor [6] is as follows:(17)$$K=\frac{\Delta R}{R}\frac{x^{2}+y^{2}(x)}{hy(x)},$$
where *K* is the sensitivity coefficient of the flexible resistance strain sensor, Δ*R* is the change in resistance value, Δ*ε* is the change in strain value, and *R* is the initial resistance value when no load is applied.

The fitted linear slope and initial resistance value of the three experimental data are substituted into Equation (17), and the analysis results of the sensitivity coefficient are shown in Table 6.

Taking the average of the three measurements,
(18)$$K=\frac{\sum_{i=1}^{n}K_{i}}{n},$$
the strain sensitivity coefficient *K* = 1.828 of the flexible resistance strain sensor was obtained.

#### 4.2.3. Uncertainty Analysis

During the calibration process, the main factors affecting the sensitivity coefficient of the flexible resistance strain sensor are the installation error of the sensor, the measurement error of the instrument, and so on. The measurement model is as follows:(19)$$K=\frac{\Delta R}{R\Delta \varepsilon },$$

The measurement uncertainty component data is shown in Table 7. Each component of the uncertainty in Table 7. The relative uncertainty of the sensitivity coefficient of the flexible resistance strain sensor measured by the nonlinear model is 0.365% (k = 2).

## 5. Discussion

The flexible strain sensor allows for multi-dimensional strain sensing, and by pasting it to the surface of the variable section cantilever beam, it is effective in avoiding the problem that transverse shrinkage occurs when the sensor is stretched. When the cantilever beam is bent, the distribution of surface strain is more uniform, which is suitable for the calibration of flexible strain sensors with a large contact area. However, the influence of the deflection angle is ignored in the theoretical derivation of the linear model. When the bending degree is small, the theoretical calculation error of strain can be ignored. However, in the case of large deflection deformation, the error value will increase with increasing deflection, and the nonlinear metrological model has an advantage in the calculation of strain in this respect. The finite element simulation revealed a difference of 200 με at both ends of the measurement area when the strain on the surface of the beam reached 5000 με. The method of calculating the surface strain by the deflection at a point on the beam is no longer applicable, so the measurement location of the deflection and the measurement location of the strain should be located in the same section. In the uncertainty analysis, the theoretical error of the model and the error introduced by the measurement are analyzed, and the influence of various factors on the calibration results is considered comprehensively. In summary, the practical application value of the research results is shown as follows: For the variable section cantilever beam calibration device, the strain calibration range is effectively improved so that it can meet the requirements of large-range calibration of flexible strain sensors. For high-sensitivity strain sensors, small errors in the theoretical strain calculation results will lead to large differences in sensitivity. The optimization of the theoretical model can effectively improve the calibration accuracy. For experimental methods, the effective range of sensor paste is determined, and the irrationality of measurement deflection at the free end is analyzed, which reduces the measurement error introduced in the experimental process. The innovation point are shown in Table 8.

## 6. Conclusions

In order to realize the high precision calibration of flexible strain sensors, an analytical strain measurement model is proposed. The model fully considers the influence of the deflection angle on the strain calculation results and combines with the finite element simulation experiment to prove the rationality of the nonlinear relationship between the deflection and strain, which effectively improves the accuracy of the strain calculation results at large deflections. The uncertainty analysis results of calibration experiments show that the theoretical error introduced by the model is the main source of uncertainty error when using the linear model, and the component is as high as 6% at 5000 με. The nonlinear model eliminates the uncertainty error component. The measurement traceability of strain is realized in the study. The nonlinear model increases the strain calibration range of a variable section cantilever beam, which provides theoretical support for the high-precision calibration of flexible strain sensors and is beneficial to promote the development and application of flexible strain sensors.

## Figures and Tables

**Figure 1 sensors-23-04778-f001:**
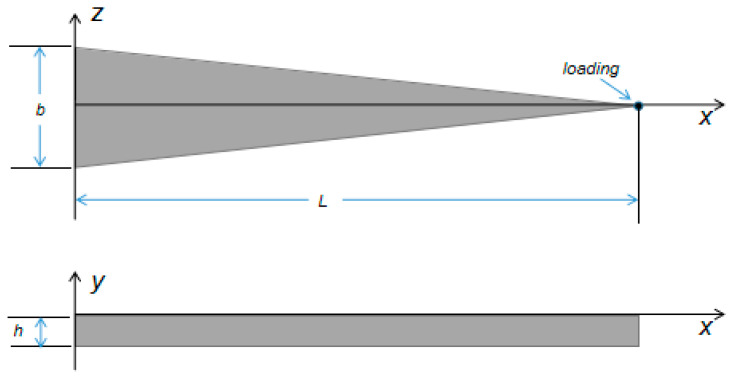
Model of a variable section cantilever beam.

**Figure 2 sensors-23-04778-f002:**
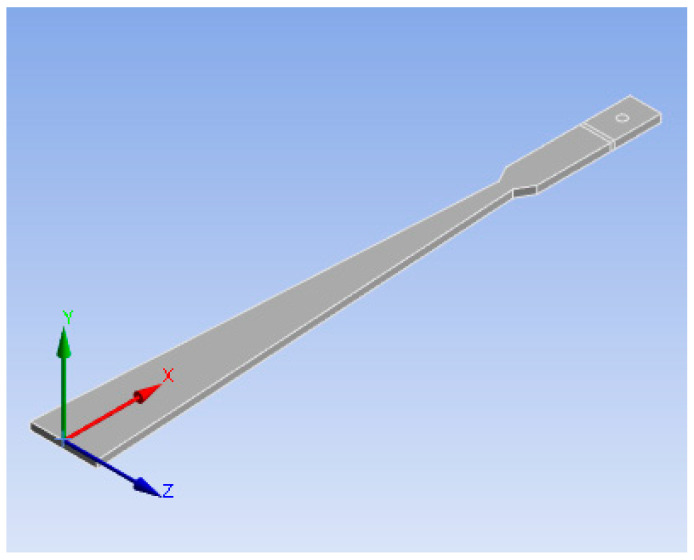
The structural model of a variable section cantilever beam.

**Figure 3 sensors-23-04778-f003:**
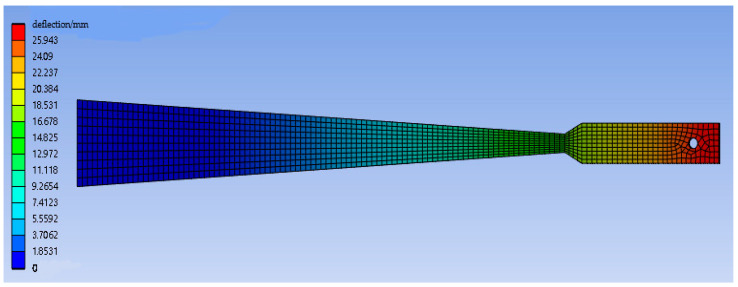
Deflection diagram of a 100 N load variable section cantilever beam.

**Figure 4 sensors-23-04778-f004:**
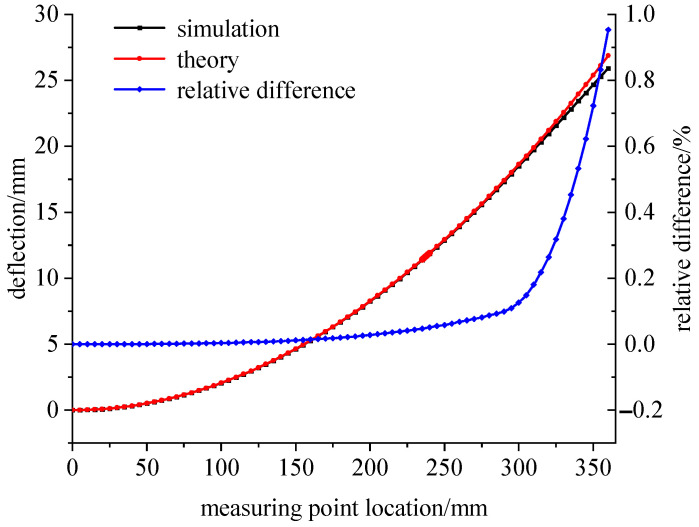
Comparison diagram of the simulation and theoretical deflections under a 100 N load.

**Figure 5 sensors-23-04778-f005:**
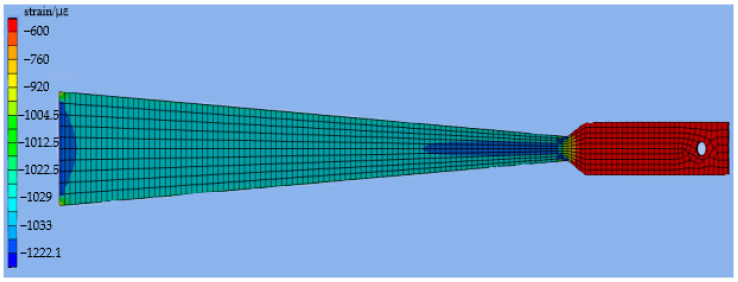
Surface strain distribution of the 100 N load.

**Figure 6 sensors-23-04778-f006:**
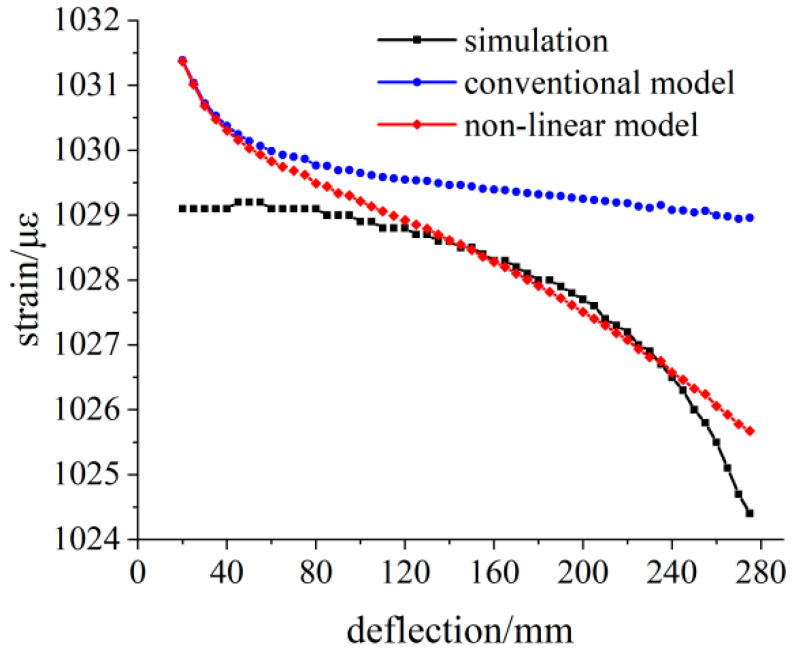
Comparison of the simulated and theoretical strains at the 100 N load.

**Figure 7 sensors-23-04778-f007:**
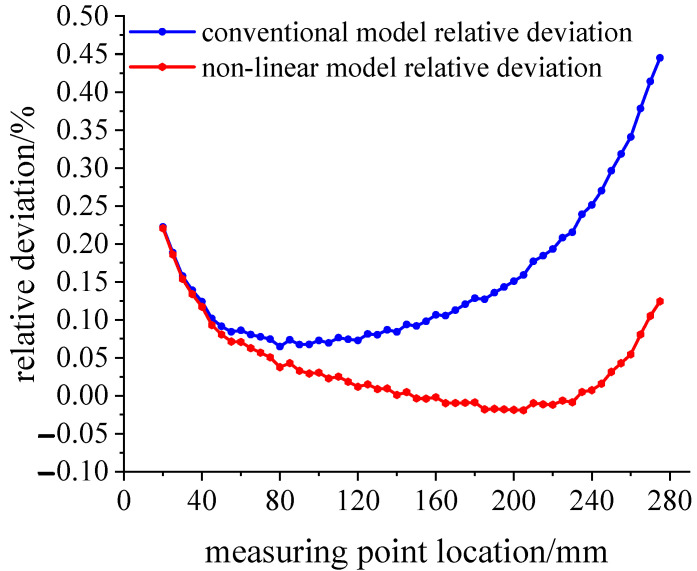
Relative deviation analysis of the theoretical strain calculation results under the 100 N load.

**Figure 8 sensors-23-04778-f008:**
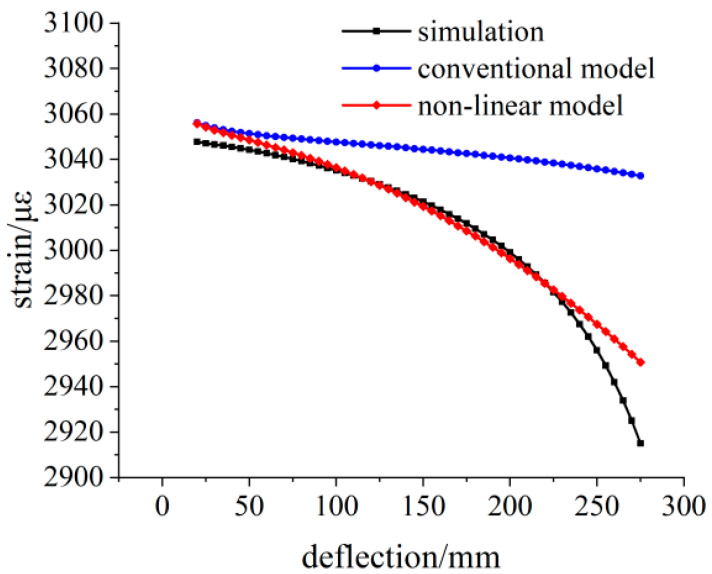
Comparison of the simulated and theoretical strains at the 300 N load.

**Figure 9 sensors-23-04778-f009:**
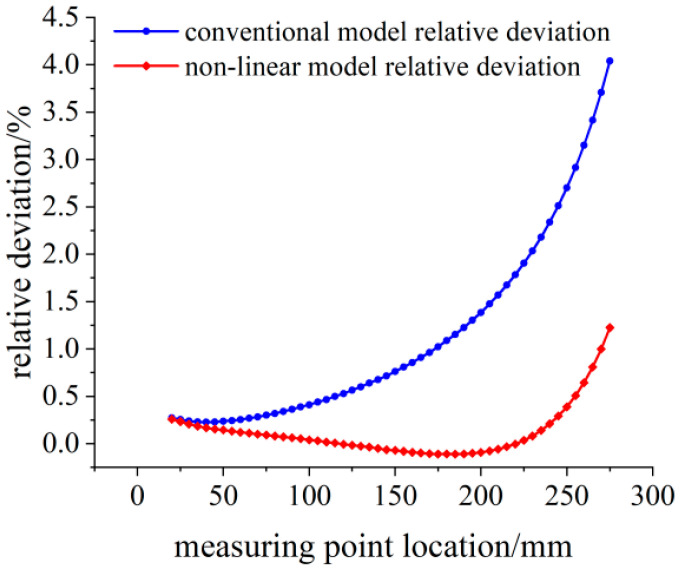
Relative deviation analysis of the theoretical strain calculation results under the 300 N load.

**Figure 10 sensors-23-04778-f010:**
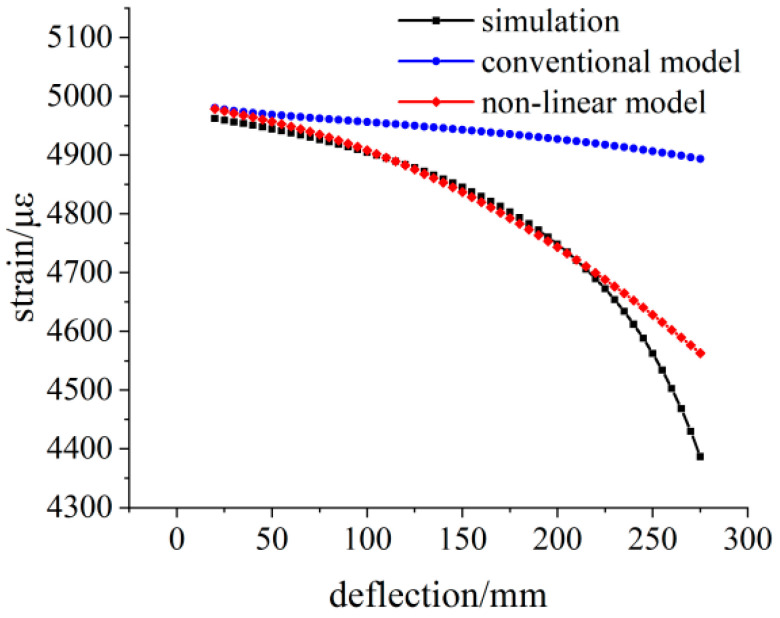
Comparison of the simulated and theoretical strains at the 500 N load.

**Figure 11 sensors-23-04778-f011:**
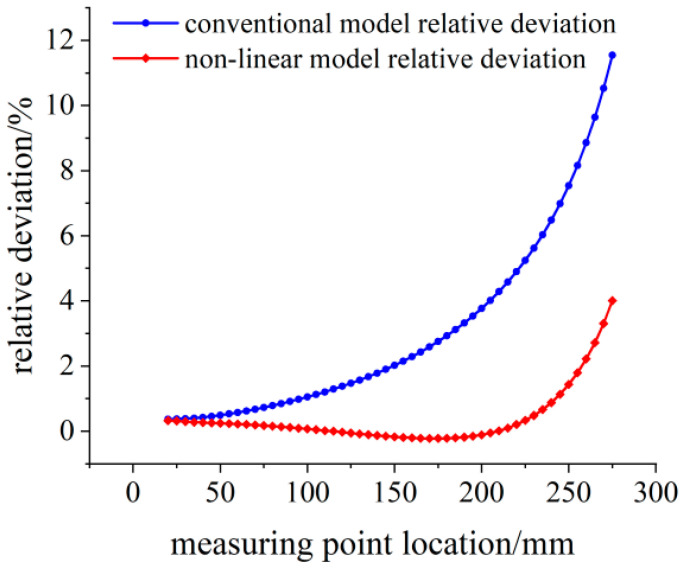
Relative deviation analysis of the theoretical strain calculation results under the 500 N load.

**Figure 12 sensors-23-04778-f012:**
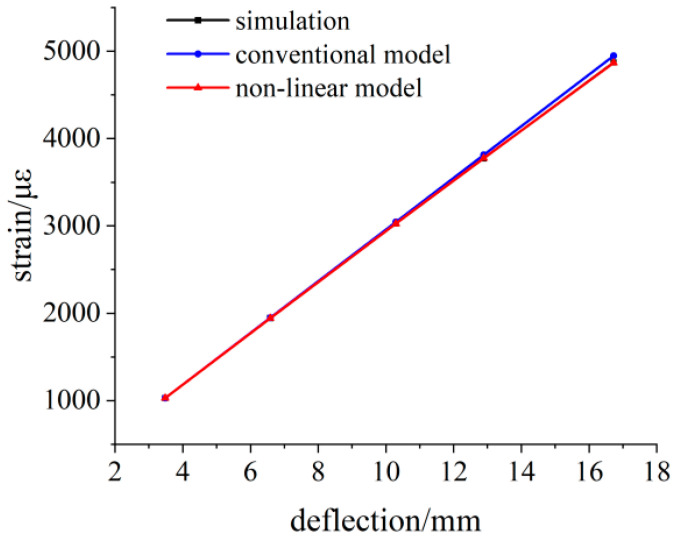
Simulation data and theoretical calculation results at 130 mm.

**Figure 13 sensors-23-04778-f013:**
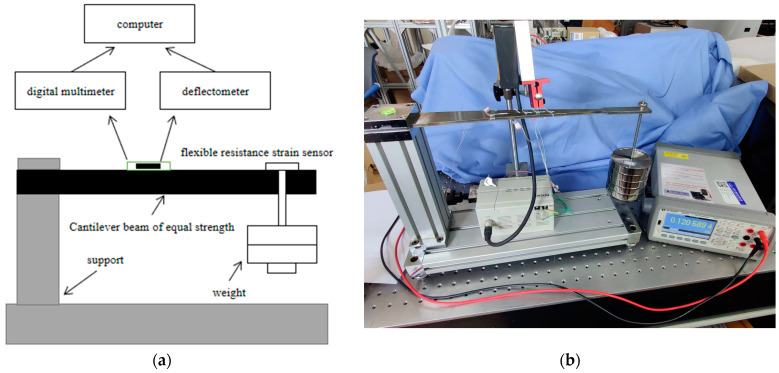
(**a**) Model structure; (**b**) strain sensor calibration device.

**Figure 14 sensors-23-04778-f014:**
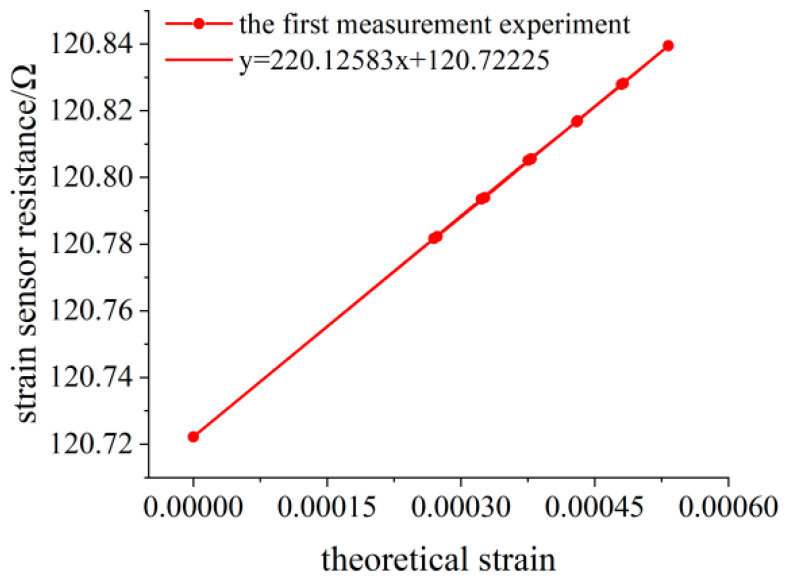
Linear fitting diagram of the experimental data for the first measurement.

**Figure 15 sensors-23-04778-f015:**
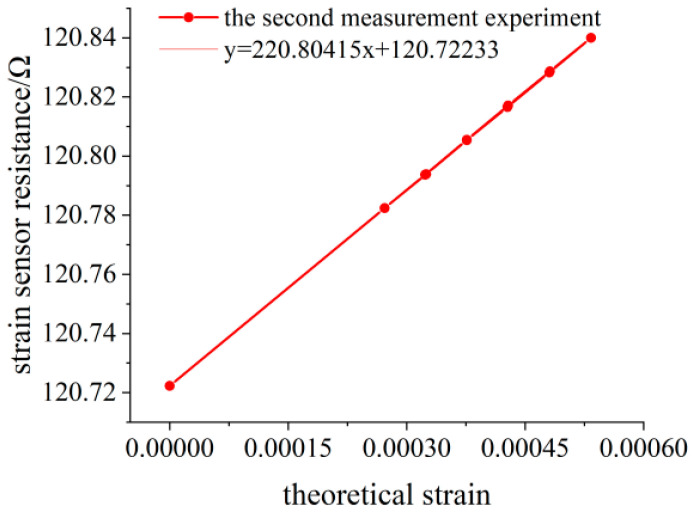
Linear fitting diagram of the experimental data for the second measurement.

**Figure 16 sensors-23-04778-f016:**
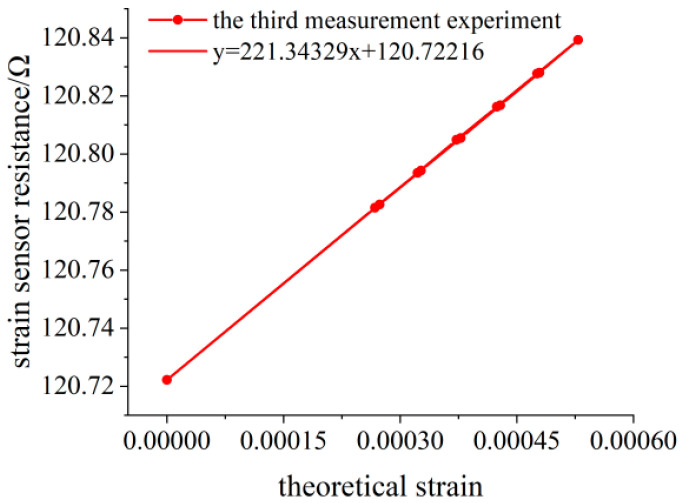
Linear fitting diagram of the experimental data for the third measurement.

**Figure 17 sensors-23-04778-f017:**
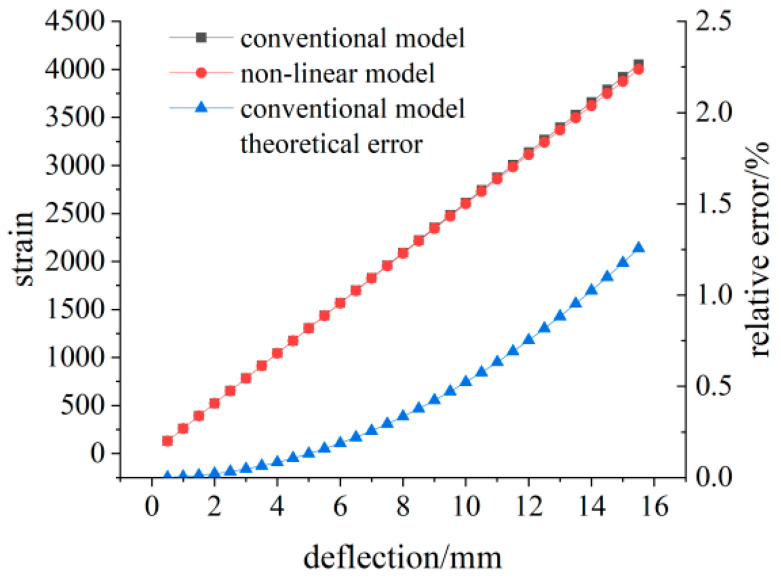
Diagram of linear model theory relative error changes.

**Table 1 sensors-23-04778-t001:** Dimension and material characteristic parameters of a variable section cantilever beam.

Parameter	Value
Materials	Spring steels of 65 Mn
Beam’s length, *L*	375 mm
Beam’s width of the fixed end, *b*	43.2 mm
Beam’s thickness, *h*	5 mm
Beam’s elasticity modulus, *E*	194 GPa
Beam’s density, *ρ*	7.85 g/cm^3^
Beam’s width, *b*(*x*)	$\{\begin{array}{cc} 0.12(360-x) & 0\leq x<285\;\mathrm{mm}\\ 1.1x-304.5 & 285\leq x<295\;\mathrm{mm}\\ 20 & 295\leq x\leq 375\;\mathrm{mm} \end{array}$

**Table 2 sensors-23-04778-t002:** Simulation data and theoretical strain calculation results at 130 mm.

Load/N	Simulation Deflection Value/mm	Simulated Strain Value/με	Theoretical Strain Values of Linear Models/με	Theoretical Strain Values of Nonlinear Models/με
100	3.4798	1028.7	1029.5	1028.8
200	6.9136	2038.8	2045.4	2039.7
300	10.295	3027.7	3045.9	3026.9
400	13.566	3972.2	4013.6	3970.4
500	16.726	4872.1	4948.5	4867.9

**Table 3 sensors-23-04778-t003:** First measurement of the experimental data.

Deflection Value/mm	Strain Sensor Resistance/Ω	Theoretical Strain Values of Nonlinear Models/με	Theoretical Strain Values of Linear Models/με
0	120.7222	0	0
1.0308	120.7817	269.45	269.46
1.2347	120.7935	322.74	322.77
1.4356	120.8051	375.24	375.28
1.6436	120.8167	429.60	429.66
1.8360	120.8279	479.87	479.95
2.0369	120.8395	532.35	532.47
1.8440	120.8282	481.96	482.04
1.6482	120.8170	430.80	430.86
1.4489	120.8056	378.72	378.76
1.2497	120.7940	326.66	326.67
1.0458	120.7823	273.37	273.38

**Table 4 sensors-23-04778-t004:** Second measurement of the experimental data.

Deflection Value/mm	Strain Sensor Resistance/Ω	Theoretical Strain Values of Nonlinear Models/με	Theoretical Strain Values of Linear Models/με
0	120.7223	0	0
1.0407	120.7824	272.04	272.05
1.2366	120.7937	323.24	323.26
1.4385	120.8055	376.00	376.04
1.6391	120.8171	428.42	428.48
1.8412	120.8287	481.23	481.31
2.0407	120.8400	533.35	533.46
1.8388	120.8283	480.60	480.68
1.6356	120.8166	427.51	427.57
1.4385	120.8054	376.00	376.04
1.2420	120.7939	324.65	324.67
1.0407	120.7824	272.04	272.05

**Table 5 sensors-23-04778-t005:** Third measurement of the experimental data.

Deflection Value/mm	Strain Sensor Resistance/Ω	Theoretical Strain Values of Nonlinear Models/με	Theoretical Strain Values of Linear Models/με
0	120.7222	0	0
1.0231	120.7815	267.44	267.45
1.2327	120.7935	322.22	322.24
1.4245	120.8049	372.34	372.38
1.6231	120.8163	424.24	424.30
1.8211	120.8277	475.98	476.06
2.0229	120.8393	528.70	528.81
1.8328	120.8280	479.03	479.17
1.6403	120.8168	428.73	428.79
1.4457	120.8055	377.88	377.92
1.2501	120.7943	326.76	326.79
1.0461	120.7826	273.45	273.46

**Table 6 sensors-23-04778-t006:** Table of the sensitivity coefficient analysis results.

	Sensitivity Coefficient
The first measurement experiment	1.823
The second measurement experiment	1.829
The third measurement experiment	1.833

**Table 7 sensors-23-04778-t007:** Summary of standard uncertainties.

Parameter	Component of Uncertainty	Classes	Relative Standard Uncertainty Component/%	Component Synthesis Standard Uncertainty/%
Δ*ε*	Thickness of variable section cantilever beam, *h*	A	0.031	0.161
	Deflection of variable section cantilever beam, *y*(*x*)	A	0.049	
	Distance from deflection measuring point to fixed end, *x*	A	0.062	
	Model’s theoretical error	A	0.085	
*R*	Initial resistance of strain gauge	A	0.009	0.009
Δ*R*	Thickness of variable section cantilever beam	A	0.085	0.085

**Table 8 sensors-23-04778-t008:** Innovation point research content.

Research Content	Existing Problem	Improvement Project
Traceability	The strain can be calculated directly through the load size, but the elastic model parameters in the formula make it impossible to realize the strain’s metrological traceability [27].	The strain is calculated by the deflection value. The parameters in the formula are displacement measurements, and the strain can be traced by measurement.
Measuring position	After the free end of the beam is widened, the deflection value is measured at the loading point to calculate the strain. The experimental results are larger than the theoretical analysis [28].	Through the finite element analysis, the free end widening makes the measured deflection greater than the theoretical value, which is consistent with the measured results in the literature. The deflection is measured in the variable section area.
Precision	The model ignores the introduction error of the deflection angle [29].	A nonlinear model is proposed to improve accuracy.

## Data Availability

Not applicable.

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
