# Peer review of "Research on a Precision Calibration Model of a Flexible Strain Sensor Based on a Variable Section Cantilever Beam"

_sensors, 2023, doi:10.3390/s23104778_

Round 1
Reviewer 1 Report
The paper presents a more accurate analytical model from a conventional one on calibrating a strain measuring system based on a variable section cantilever beam. The models are benchmarked using numerical simulations. The maximum value of relative error of estimated strain between non-linear and conventional model is 1,3% at 4000 με that is the range of operation of flexible strain sensors (fig.17). The following points need to be addressed before considering publication:
1) Is the strain measuring cantilever beam system a commercial one or a home made system? If yes, please provide information.
2) The final relative correction after the application of the non-linear model being quite small, the authors should discuss the importance of their findings in practical applications.
3) From fig.17 the relative maximum correction is 1,3% at 4000 με. In the abstract it is reported a 6% correction for 5000 με. Is that value correct ? Can you give information about its estimation ?
4) What kind of flexible resistive strain sensor is used? Is it a commercial one? If yes, please give the reference.
5) The strain sensor tested gives an average sensitivity at 1,828. Could you discuss the influence of the nonlinear model for higher sensitivity (f.e. 100) sensors?
English language used in the manuscript needs further polishing.
Reviewer 2 Report
Sensors- 2387884
Title: Research on precision calibration model of flexible strain sensor based on variable section cantilever beam.
Indeed, the manuscript is well-written and easy to follow. Some points need to be known.
-It will be good to write the reference for equation 1 etc.
-In Table 2, the maximum difference is 4.2 με, as reported. What may be the source of error for this difference?
- Can authors compare the experimental results with simulation results for different types of loads (100N, 200N etc.)
-It will be good to include the real picture of the experimental setup along with Figure 13.
-Please explain Table 5 in more detail.
-Write the specifications of the LiDAR IC used in this work.
-Have the authors investigated the proposed strain calibration device for different values of temperature?
-The novelty of the work should be clearly highlighted (in the abstract and the conclusions).
-It is better to list a comparison table to compare results with previous work.
-More latest references should be added.
Moderate editing of English language
Round 2
Reviewer 1 Report
The authors have satisfactorily replied to this reviewer's comments and the paper can be published.
Author Response
Thank you again for your suggestions on the revision of the manuscript.
Reviewer 2 Report
sensors-2387884
Research on precision calibration model of flexible strain sensor based on variable section cantilever beam
Thank you for allowing me to revise resubmitted manuscript titled " Research on precision calibration model of flexible strain sensor based on variable section cantilever beam" I believe the submitted manuscript and presented work is suitable for publishing in sensors, except for one minor revision.
Please explain the possible sources of errors more specifically, as 6% error is mentioned in the manuscript.
Moderate editing of English language
